# Deep Transition-Encoding Networks for Learning Dynamics

**David van Dijk[†], Scott Gigante[†], Alexander Strzalkowski, Guy Wolf[‡] & Smita Krishnaswamy[‡]\* \***
Yale University
New Haven, CT 06520, USA
`smita.krishnaswamy@yale.edu`

## Abstract

Markov processes, both classical and higher order, are often used to model dynamic processes, such as stock prices, molecular dynamics, and Monte Carlo methods. Previous works have shown that an autoencoder can be formulated as a specific type of Markov chain. Here, we propose a generative neural network known as a transition encoder, or *transcoder*, which learns such continuous-state dynamic processes. We show that the transcoder is able to learn both deterministic and stochastic dynamic processes on several systems. We explore a number of applications of the transcoder including generating unseen trajectories and examining the propensity for chaos in a dynamic system. Finally, we show that the transcoder can speed up Markov Chain Monte Carlo (MCMC) sampling to a convergent distribution by training it to make several steps at a time.

## 1 Introduction

Since the popularization of the autoencoder in 2006 (Hinton & Salakhutdinov, 2006), they have been used for denoising (Vincent et al., 2008; Rifai et al., 2011), data generation (Kingma & Welling, 2013) Bengio et al. (2013) Alain et al. (2015) and data visualization (Amodio et al., 2017). However, the general framework of an autoencoder is amenable to a wide variety of applications. Here, we reinterpret and generalize an autoencoder as a *transition encoder*, or transcoder, which is capable of learning dynamic systems. First, we note that an autoencoder is essentially a first-order Markov chain that walks towards the data manifold. Regularizations and bottlenecks including low-dimensional hidden layers, input noise or L1/L2 regularization do not allow for the exact recreation of the input, but rather can be seen as a transitional step towards the data manifold. Generalizing this notion in several ways, we train a neural network on samples from a stochastic dynamic system, where the network learns not to recreate its own input but instead to transition to one of the possible next states of the system, thus learning a deep embodiment of an $n$th order Markov process.

The advantages of learning a neural network to produce a stochastic transition are two-fold. First, the transcoder is generative and can produce new trajectories through the system when sampled in a chain-like fashion (with the output fed back to the network as the input of the next cycle). Such generated trajectories can be studied further, for example to assess low energy states and propensity for chaos. Second, since the transcoder itself is an embodiment of the transition logic, its hidden layers can be used to visualize the energy landscape of the system.

The main contributions of this manuscript are:

1. Reinterpretation of existing autoencoders as deterministic Markov chains;
2. Formulation of the transcoder architecture which is capable of learning any deterministic or stochastic $n$th order Markov process;
3. Application of the transcoder to learning and sampling from several types of dynamic systems including test cases such as a single and double pendulum;
4. Acceleration of traditional Markov Chain Monte Carlo sampling by "fast-forward" transcoder training.

---

*[\*†] These authors contributed equally. [‡] These authors contributed equally.*

Like an autoencoder, the input and output layers of the transcoder have the same dimensions. However, instead of training to recreate the input, we train to map the input to an adjacent output, based on a transition function or sampling from a dynamical system. In the deterministic case, we map each input to a single output. To enable stochastic dynamics, the stochastic transcoder takes an extra noise input and learns to transmute this noise input into the conditional distribution of potential transitions for points in the input space. Thus, to train the transcoder against the appropriate conditional distributions for each point, we use a probabilistic Maximum Mean Discrepancy (MMD) loss first used in Gretton et al. (2012) Dziugaite et al. (2015). Additionally, we build a degree of invariance to the input to effectively interpolate between learned input states and function in continuous state spaces where it is impossible to see all inputs. This is achieved by adding corruption noise to the input in addition to the side-noise input. Thus, the transcoder can learn stochastic dynamic transitions in continuous state spaces.

## 2 EMPIRICAL RESULTS

In this section, we demonstrate the performance of the transcoder on a variety of datasets to show the capabilities of the deterministic, stochastic and $n$th order transcoder. We show that the transcoder is able to learn processes that are harmonic or chaotic, ergodic or null-recurrent. We further show that the transcoder can effectively sample from a dynamic process faster than is possible using classical methods such as Markov Chain Monte Carlo.

### 2.1 HARMONIC AND CHAOTIC SYSTEMS

We train the $n$th order transcoder on two deterministic systems, a single and double pendulum. In the single pendulum, we use a second-order transcoder and in the double pendulum, we use a third-order transcoder in each case using the angles of the pendulums as network inputs. Figure 2.1 shows the Euclidean coordinates of the transcoder-generated paths of both pendulums over time, with only the second pendulum shown in the case of the double pendulum. Both pendulums show smooth trajectories, with the single pendulum showing periodic behavior and the double pendulum showing chaotic behavior.

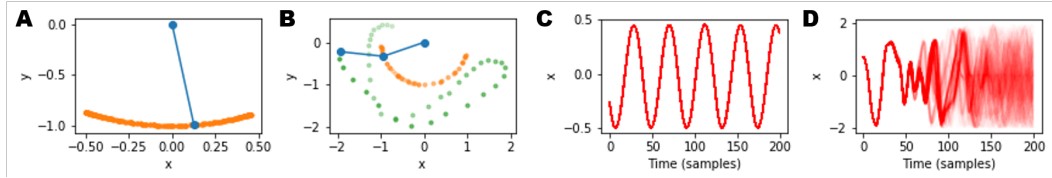

Figure 2.1: A, B. Single paths generated by the second-order transcoder with a single (A) and double pendulum (B). C, D. X coordinate of 500 paths on single (C) and double (D) pendulums, starting from an epsilon-difference, where the lower of the two pendulums is shown in the double pendulum. Videos of both pendulums can be viewed at http://bit.do/transcoder.

### 2.2 STOCHASTIC SYSTEMS

We train the stochastic transcoder on the Frey faces dataset generated by Roweis & Saul (2000). The input to the network given as intensity values for 560 pixels (28x20, grayscale) and a single Gaussian noise input. Training examples are drawn from a single time series of input from a video, with transitions sampled from $(x_t, x_{t+n}), -12 < n < 12$, where $x_t$ is the $t$th frame of the video. Figure 2.2 shows 1000 samples generated by the stochastic transcoder on Frey faces dataset. Training data is shown in gray. The first 1000 samples are discarded. Ten samples are drawn at uniform spacing from the Frey faces path and displayed below the PCA embedding. The transcoder samples a large range of states lying on the data manifold and creates an new trajectory through a space.

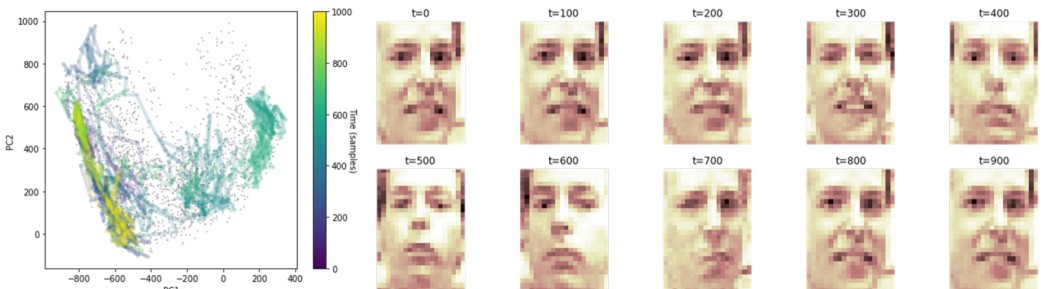

Figure 2.2: Chain generated by the stochastic transcoder visualized on a PCA embedding (left) of Frey faces dataset. The chain is shown in color superimposed over the training data in gray with color indicating time. 10 equally spaced faces from the Frey faces chain generated by the transcoder are shown at right. A video of the Frey faces chain can be viewed at http://bit.do/transcoder.

## 2.3 FAST-FORWARDING MCMC

We train the stochastic transcoder to sample a Gaussian mixture model with the goal of approximating Markov Chain Monte Carlo sampling. We generate training data from a 1-dimensional Gaussian mixture model using Metropolis-Hastings sampling, with eight independent chains sampled for 10,000 samples after 4,000 samples of burn-in, where samples are proposed to the MCMC algorithm by applying Gaussian noise with standard deviation 0.1 to the current sample. We draw training samples from the generated chains after density subsampling and are provided to the transcoder as $(x_t, x_{t+1})$.

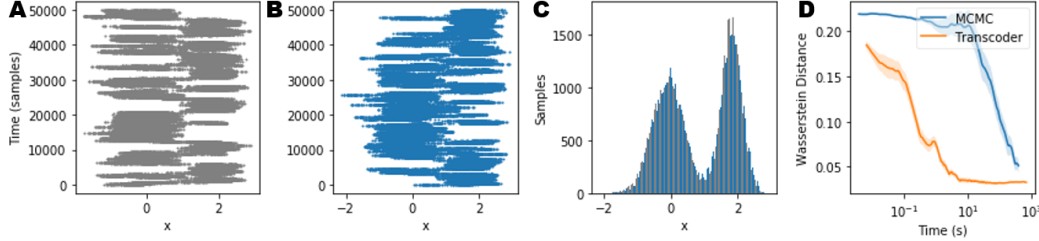

Figure 2.3: A, B. Samples generated from a Gaussian mixture model with Metropolis-Hastings (A) and by the stochastic transcoder (B). C. Marginal distributions of both paths. D. Wasserstein distance between ideal Gaussian mixture model and chains sampled from MCMC and fast-forwarded transcoders over time. MCMC is sampled with step size 0.01, and the transcoder is trained to skip 300 steps in this MCMC process. Standard error of the mean is shown (n=32).

Figure 2.3 shows a single chain of 20,000 samples from Metropolis-Hastings and transcoder sampling respectively. The transcoder emulates both the dynamics and the overall distribution of the MCMC sampling. We extend this example to demonstrate the ability of a transcoder to outperform MCMC sampling by training the same transcoder on Metropolis-Hastings generated chains this time generated with sampling noise of standard deviation 0.01, where training examples are provided to the transcoder as $(x_t, x_{t+300})$ such that the transcoder learns to "fast-forward" and skip steps of the MCMC sampling procedure. Figure 2.3D shows the convergence of a sampled chain to the theoretical distribution over time, where the transcoder is able to sample a representative chain several orders of magnitude faster than MCMC. The transcoder was trained for 550 seconds with 2617MB of RAM on a NVIDIA GeForce Titan X Pascal GPU.

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
