# OpenReview forum: "Deep Transition-Encoding Networks for Learning Dynamics"
_ICLR.cc/2018/Workshop — Reject_

### Official Review · AnonReviewer3 · 2018-02-21
**Lazy cut down of longer paper**

**Rating:** 2
**Confidence:** 4

**Review:**

This submission is effectively a lazy cut down of the preprint

https://arxiv.org/pdf/1802.03497.pdf?fname=cm&font=TypeI

where the authors have simply taken the introduction and some of the empirical results (without any noticeable rewriting) and hoped that these still make sense with most of the paper removed.  Unsurprisingly, this is not the case and results in a somewhat nonsensical submission without any clear technical content.  The method is left almost completely unexplained and the technical contribution is very unclear.  As such it is very difficult to argue for acceptance.

Specific points:
- The "reinterpretation of existing autoencoders as deterministic Markov chains" is completely unclear.  I have no idea what you mean by this other than the obvious and uninteresting fact that both are mappings from an input space to an output space.
- Monte Carlo methods are neither a dynamic process nor something that you model.  I understand that what you mean is that stochastic Markov processes are used for inference via MCMC, but what you actually say is quite different to this.
- The transcoder will clearly convergence to the wrong distribution for the MCMC context (see e.g. figure 2.3D) and so the last sentence of the abstract is at best misleading and worst straight up wrong.  Same goes for the last sentence of the first paragraph of section 2.
- The concept of fast-forwarding MCMC is itself rather spurious, or at least a very convoluted view of the problem.  An MCMC sampler that carries out multiple steps in one go is itself an MCMC sampler, just one with a better proposal.  Thus what you are really doing here is proposal adaptation, something which is well established in the Bayesian inference literature (see e.g. Approximate Inference with Amortised MCMC by Li et al 2017, for some recent work in the area).

---

### Official Review · AnonReviewer2 · 2018-03-14
**Unclear presentation, contribution, and results.**

**Rating:** 3
**Confidence:** 4

**Review:**

This paper seems to be very early work on reinterpreting autoencoders as Markov chains. This idea by itself is not new, and has been explored extensively in the literature, and it's not clear from the presentation if there is anything new the authors want to add to the conversation.

It's not at all clear what the algorithm/model actually is, since the authors chose to use an imprecise language (English) to describe it, instead of mathematical notation or an algorithm box. A reader unfamiliar with the literature would not gain anything at all from reading since they wouldn't be able to appreciate what the algorithm even is.

Even if the reader manages to deduce the algorithm (using their prior expertise), the results are not presented well. It's not clear what to expect, and what the authors are trying to show.

---

### Decision · Program_Chairs · 2018-03-20
**ICLR 2018 Workshop Acceptance Decision**

**Decision:**

Reject

**Comment:**

Based on the reviews, this paper has not been accepted for presentation at the ICLR workshop. However, the conversation and updates can continue to appear here on OpenReview.